# Consensus of Multi-Agent Systems with Unbounded Time-Varying Delays

**Siheng Zong** [1,*] and **Yu-Ping Tian** [2,*]

1 School of Automation, Southeast University, Nanjing 210096, China
2 School of Automation, Hangzhou Dianzi University, Hangzhou 310018, China
* Correspondence: shzong@seu.edu.cn (S.Z.); yptian@hdu.edu.cn (Y.-P.T.)

**Abstract:** In multi-agent systems with increasing communication distances, the communication delay is time-varying and unbounded. In this paper, we describe the multi-agent system with increasing communication distances as the discrete-time system with non-distributed unbounded time-varying delays and study the consensus problem of the system via the distributed control. This paper uses a time-delay system to model the discrete-time system, and the maximum delay in the time-delay system tends to infinity as time goes on. Furthermore, caused by this property, most of convergence analysis methods for bounded time-delay systems are ineffective. Hence, for any finite integer $k > 0$, the finite-dimensional augmented model of the time-delay system is built in the interval $[0, k]$ to study the system state. Under the weaker topological assumption that the topology containing a spanning tree, the system is proved to achieve a consensus if the growth rate of the maximum delay satisfies some mild constraints, which also are constraints on the growth rate of the maximum communication distance between agents. Furthermore, we characterize that the rate of the system achieving a consensus and the growth rate of the maximum delay are negatively correlated. In other words, the rate of the system achieving a consensus and the growth rate of the maximum communication distance between agents are negatively correlated.

**Keywords:** consensus; convergence rate; discrete-time system; increasing communication distances; Infinite maximum delay; time-delay system; unbounded time delay



## 1. Introduction

The multi-agent system has become a very popular topic in the control community in recent years. This is mainly due to its broad applications in practical systems, such as formation control (see [1,2]), time synchronization (see [3–5]), etc. In the study of the multi-agent system, a fundamental problem is to design a distributed controller such that the states or the outputs of all agents reach an agreement whereas every agent has only access to the information of its neighboring agents. Such a problem, which is called the consensus problem, has been studied widely (see [6,7]).

It is a well-known fact that delay is unavoidable in many practical systems, which can lead to oscillatory instability, chaos and bifurcation (see [8–10]). In the multi-agent system, many factors such as the packet loss, the channel congestion, and the communication distance can cause delays. When the number of agents is large and the distance between these agents is small, the delay caused by the communication distance is very small and is often ignored. However, in order to perform precise tasks, the state between agents needs to be exactly synchronized. For example, in the field of the military reconnaissance, if a distributed multi-agent system needs to cooperatively observe targets, the agents need to achieve exact time synchronization. In this type of system, the time synchronization error between agents needs to reach the microsecond or nanosecond level (see [11]), and the smaller the time synchronization error is, the better the cooperative task of agents is completed. In this case, although the delay caused by the communication distance is

very small, which may only reach the millisecond or microsecond level, it still cannot be ignored. Furthermore, when the number of agents in the system is small and the distance between agents is far, the density of the network is small. In this kind of network, the delay caused by the packet loss and the channel congestion decreases (see [12]). On the contrary, the delay caused by the communication distance increases and has a greater impact on the communication time-delay. Therefore, this paper mainly studies the influence of the time-delay caused by the communication distance on the consensus of the distributed multi-agent system.

Generally, time delays studied in multi-agent systems can be divided into constant delays (see [13,14]), time-varying delays (see [15–17]) and random delays (see [18]). In most of the researches, the above-mentioned delays were assumed to be bounded. To solve the consensus problem of multi-agent systems with bounded time-delays, researchers use the following common methods. In [15,18], by expanding the dimension of the system by a finite multiple, the multi-agent system with bounded time-delays is transformed into the multi-agent system without time-delay. Furthermore, the Lyapunov function is used to solve the consensus problem. In [14,16], the consensus problem of the system is transformed into a stability problem, and then the Lyapunov–Krasovskii functional with double integral is used to get conditions to ensure that the derivative of the Lyapunov–Krasovskii functional is negative definite. These conditions are related to time-delays and are sufficient conditions to ensure the stability of the system. In the Lyapunov–Krasovskii functional, since integration limits of the double integral are related to time-delays, the derivative of the Lyapunov–Krasovskii functional with double integral contains a positive quadratic term whose coefficient is the time-delay. The boundedness of the time-delay guarantees the existence of conditions to ensure that the derivative of the Lyapunov–Krasovskii functional is negative definite. In [13,17], under the discrete-time multi-agent system, after the consensus problem of the system is transformed into a stability problem, the discrete Lyapunov–Krasovskii functional with double summation is used to get stability conditions related to time-delays. In the difference of the discrete Lyapunov–Krasovskii functional, there is a positive quadratic term and its coefficient is the time-delay. Therefore, only if the time-delay is bounded, conditions that makes the difference negative definite exist.

In fact, the time delay may be unbounded. When we use a small number of nodes for search and rescue work in a large area, the communication distance between agents keeps increasing. For the multi-agent system with increasing communication distances, since the information transmission time and the communication distance are positively correlated, the time delay tends to infinity as time goes on. Hence, the multi-agent system with increasing communication distances is described as the multi-agent system with unbounded time-delays. Due to the unboundedness of delays, the above-mentioned method no longer applies. With unbounded time-delays, a finite-dimensional extended system cannot describe the multi-agent system with unbounded time-delays at all time. Hence, methods used in [15,18] are ineffective. Then, let us consider the Lyapunov–Krasovskii functional with double integral such as [14,16]. With unbounded time-delays, the positive quadratic term whose coefficient is the time-delay tends to positive infinity as time goes on. Therefore, the derivative of the Lyapunov–Krasovskii functional can not be negative definite. Furthermore, under the discrete-time multi-agent system, the discrete Lyapunov–Krasovskii functional with double summation faces the same problem. Hence, the dynamical study of unbounded delay system is recognized as being very difficult. The unbounded delay systems that have received attention are summarized into two categories.

The first category is the multi-agent system with distributed infinite delays. Many practical systems, such as the traffic flow (see [19,20]) and biological networks (see [21]), can be modeled as the system. In this category, the weighted delay information is used to control the system, and the greater the weight of the delay information is, the smaller the delay is. By using the algebraic graph theory and frequency domain analysis, Reference [19] shows that with the gamma distribution weight, the system can achieve a consensus if

the topology is strongly connected. Further, under a weaker topological condition that the topology contains a spanning tree, Reference [20] shows that with the exponential distribution weight, the system can achieve a consensus by using a low gain controller. It can be noticed that, in this category, agents need to know the accurate delay of each received information. However, it is difficult to achieve in practical systems. In order to avoid this weakness, the second category is considered.

The second category is the multi-agent system with non-distributed unbounded time-delays. Some systems with a spacial nature, such as neural networks (see [22–24]), can be modeled as the system. Different from the first category, in the second category of unbounded delay system, the delay of each received information is unknown to any agent. For the second category, many results are about the stability of the system (see [22–24]). In [22], the authors introduced generalized Halanay inequalities to study the stability of the system. In [23], the method, which is based on upper bounding of the state vector by a decreasing function, is presented to analyse the stability of the system. In [24], based on an impulsive differential delay inequality, several novel delay-dependent inequalities are obtained to ensure the global stability of the system. However, only a few results are about the consensus of the system (see [25,26]). When the adjacency matrix of the topology is impartible, by using the algebraic graph theory and time domain analysis, the system achieves a consensus (see [25]). Furthermore, under the topology that satisfies the spanning tree assumption and the no-cycle assumption, Reference [26] solves the consensus problem of the system. However, to make the system achieving a consensus, assumptions of topologies in these results are quite strict.

In this paper, the multi-agent system with increasing communication distances is described as the discrete-time distributed system with non-distributed unbounded time-delays. The first goal of this paper is to use distributed algorithms to make the discrete-time system achieving a consensus under a weaker topology assumption. This paper uses a time-delay system to model the discrete-time system, and the maximum delay in the time-delay system tends to infinity as time goes on. Then, the consensus problem of the time-delay system is studied through a new method. It can be noticed that, for any finite integer $k > 0$, the maximum delay is bounded in the interval $[0, k]$. Hence, for any finite integer $k > 0$, the finite-dimensional augmented model of the time-delay system can be built in the interval $[0, k]$. Furthermore, these finite dimensional system models are used to study the system state. By using this method, under the fixed directed topology containing a spanning tree, which is weaker than topology assumptions in existing results (see [25,26]), the system is proved to achieve a consensus, if a mild assumption satisfied by the rate of the maximum delay tending to infinity. Since the maximum delay and the maximum communication distance are positively correlated, the assumption of the growth rate of the maximum delay also is the constraint on the growth rate of the maximum communication distance. The second goal of this paper is to study the convergent rate of the system. This paper reveals a new feature that the rate of the system achieving a consensus and the growth rate of the maximum delay are negatively correlated. In other words, rate of the system achieving a consensus and the growth rate of the communication distances between agents are negatively correlated. These theoretical results are verified by simulation results. Furthermore, these results are not only applicable to the multi-agent system with increasing communication distances, but also to any multi-agent system that can be described as the multi-agent system with non-distributed unbounded time-varying delays.

The rest of this paper is organized as follows. In the second section, the time-delay system model of the multi-agent system with increasing communication distances is introduced. Furthermore, for any integer $k \geq 0$, the finite-dimensional augmented model of the time-delay system in interval $[0, k]$ is built. In the third section, conditions and the rate of the system achieving a consensus is studied. The fourth section uses numerical simulations to verify theoretical results given in the paper.

Basic symbols and concepts: $R$ means the set of real number, and $Z$ means the set of integers. $R^{m \times n}$ means $m \times n$-dimensional real space. $0_{m \times n}$ means the $m \times n$-dimensional $0$

matrix. The unit matrix of an appropriate dimension is represented by $I$. For any matrix $A$, the $i - j$ entry of the matrix is represented by $A_{ij}$. Let $\Lambda(A) = \{A' | A'_{ij} = A_{ij} \text{ or } 0\}$. For any vector $x \in R^n$, the $i$th entry of $x$ is represented by $x_i$. For functions $f(x), g(x), x \in D$, where $D$ is the definition domain, $f(x) = O(g(x))$ means that there exists a positive real number $c_1$ such that $f(x) \leq c_1 g(x)$ for all $x \in D$. For an $x_0 \in D$, $f(x) = o(g(x))$ as $x \to x_0$ means that $\frac{f(x)}{g(x)} \to 0$ as $x \to x_0$, where if the definition domain $D$ is unbounded, $x_0$ could be infinity. Furthermore, $f^{-1}(x)$ means the inverse function of $f(x)$. A function $f(x)$ is called as a $\mathcal{K}_\infty$ class function, if $f : [0, +\infty) \mapsto [0, +\infty)$, $f(0) = 0$ and $f(x)$ is strictly increasing. $\max_j\{\cdot\}$ and $\min_j\{\cdot\}$ represent maximum and minimum functions, respectively. For $a \in R$ and $b \in Z$, $\lfloor a \rfloor = \max\{b | b \leq a\}$. For a set $C$, $|C|$ means the number of elements of the set $C$.

For a directed graph (digraph) $G = (V, E)$, $V$ represents agent set and $E$ represents edge set. Furthermore, if there is a directed edge from the agent $i$ to the agent $j$, then $(i, j) \in E$. A directed path from the agent $i$ to the agent $j$ is made up by a sequence of directed edges in the edge set $E$ connected end to end: $(i, i_1), (i_1, i_2), \ldots, (i_{k-1}, i_k), (i_k, j)$. Furthermore, the length of this directed path is the number of directed edges that make up this directed path, i.e., $k + 1$. $\text{dis}(i, j)$ represents the distance from the agent $i$ to the agent $j$, which is the length of the shortest directed path from the agent $i$ to the agent $j$.

## 2. Discrete-Time Distributed System and Its Mathematical Model

In the multi-agent system with increasing communication distances, each agent updates its state and sends the state information to its neighbors according to the period $T$. If an agent does not receive the state information sent by one of its neighbors in a period $T$, then the agent updates its own state with the last received state information of the neighbor. Consider the digraph $G = (V, E)$ with $n$ agents, where $V = \{1, 2, \ldots, n\}$, $E$ is the set of all directed links in digraph $G$. Furthermore, if $(i, j) \in E$, agent $j$ can receive the information sent by agent $i$. In this paper, we have the following assumption on digraph $G$.

**Assumption 1.** *The digraph $G$ contains a spanning tree.*

As we all know, the communication delay and the communication distance are positively correlated. In this paper, we take the approximately linear relationship between the time delay and the communication distance as an example, and we have $m_{ji}(k) = \lfloor bC_{ji}(k) \rfloor$, where $b > 0$ is a constant, the integer $m_{ji}(k) \in [0, k]$ is the delay in the directed link $(i, j)$, and $C_{ji}(k)$ is the communication distance between agents $i$ and $j$. The consensus algorithm of the multi-agent system with increasing communication distances can be described as the discrete-time distributed system with non-distributed unbounded time-delays:

$$x_j(k+1) = x_j(k) + c \sum_{i=1}^{n} a_{ji}\big(x_i(k - m_{ji}(k)) - x_j(k)\big) \tag{1}$$

for all $j \in V$, where $c \in (0, 1)$ is a designable constant gain of system (1), and $x_j \in R$ is the state of agent $j$. In (1), $\sum_{i=1}^{n} a_{ji} = 1$, where $a_{ji} > 0$ when $(i, j) \in E$ and $a_{ji} = 0$ otherwise. For any $j \in V$, the initial state $x_j(0)$ is given. For the multi-agent system (1) with increasing communication distances, the following assumption means that the communication distance between agents mast increase slower than the speed of information transfer. In other words, for any $(i, j) \in E$, the following assumption guarantees that the agent $j$ can receive the information sent by the agent $i$.

**Assumption 2.** *For any $(i, j) \in E$ and any integer $k \geq 0$, $m_{ji}(k) \in Z \bigcap [0, k]$ and $k - m_{ji}(k) \to \infty$ as $k \to \infty$.*

When the following definition is satisfied, system (1) is called achieving a consensus.

**Definition 1.** *System* (1) *achieves a consensus, if there exists a constant* $x^*$ *such that*

$$|x_j(k) - x^*| \to 0, k \to \infty \tag{2}$$

*for all* $j \in V$.

Let $M(k)$ be the maximum delay, i.e., $M(k) = \max_{\{s \in Z \cap [0,k] \text{ and } i,j \in V\}} \{m_{ji}(s)\}$. Assumption 2 shows that the maximum delay $M(k)$ could be unbounded. In other words, when $k \to \infty$, $M(k) \to \infty$. Then, for all integer $k \in [0, \infty)$, the system (1) cannot be described by one finite-dimensional time-delay system. To solve this problem in the modelling and consensus analysis, for any finite integer $K > 0$, system states are studied by building a finite-dimensional augmented model in the interval $[0, K]$. Let $x(k) = [x_1(k), \dots, x_n(k)]^T$ for all integer $k \in [0, \infty)$. Without loss of generality, let $x(k) \equiv x(0)$ for all $k \in \{0, -1, \dots\}$. Furthermore, for any integer $K \in [0, \infty)$ and integer $k \in [0, K]$, let $y_{x,K}(k) = [x(k)^T, x(k-1)^T, \dots, x(k - M(K))^T]^T$. Then, for any integer $K \in [1, \infty)$ and all integers $k \in [0, K-1]$, system (1) is equivalent to

$$y_{x,K}(k+1) = A_K(k)y_{x,K}(k), \tag{3}$$

where

$$A_K(k) = \begin{bmatrix} \widehat{A_{K0}}(k) & \widehat{A_{K1}}(k) & \cdots & \widehat{A_{KM(K)-1}}(k) & \widehat{A_{KM(K)}}(k) \\ I & 0_{n \times n} & \cdots & 0_{n \times n} & 0_{n \times n} \\ 0_{n \times n} & I & \cdots & 0_{n \times n} & 0_{n \times n} \\ \vdots & \vdots & \ddots & \vdots & \vdots \\ 0_{n \times n} & 0_{n \times n} & \cdots & I & 0_{n \times n} \end{bmatrix}. \tag{4}$$

The Laplacian matrix of the digraph $G$ is represented by $L$, which satisfies that for any $i, j \in V$, if $i \neq j$ then $L_{ji} = -a_{ji}$; for all $j \in V$, $L_{jj} = 1$. Let $\check{A} = I - cL$. In (4), the matrix $A_K(k)$ satisfies properties that $\widehat{A_{Ks}}(k) \in R^{n \times n}$ and $\widehat{A_{Ks}}(k) \in \Lambda(\check{A})$ for all $s \in \{0, \dots, M(K)\}$. Furthermore, we have $\sum_{s=0}^{M(K)} \widehat{A_{Ks}}(k) = \check{A}$. Diagonal entries of $\widehat{A_{K0}}(k)$ are the same as diagonal entries of $\check{A}$. For any $i, j \in V$, if $i \neq j$ then $\widehat{A_{Km_{ji}(k)}}(k)_{ji} = \check{A}_{ji}$. For any integer $K > 0$, the set of matrices $A_K(k)$ for all $k \in Z \cap [0, K-1]$ is represented by $S(K)$, and the number of different matrices in $S(K)$ is $|E|(M(K)+1)$.

## 3. Convergence Analysis

This section studies the consensus problem of system (1). Firstly, under Assumption 2, properties of system states in the interval $[0, K]$ for any finite integer $K > 0$ is studied. Let $\tilde{c} = \min\{\min_{i,j \in V}\{a_{ij}|a_{ij} \neq 0\}c, 1-c\} \in (0,1)$, $D_G \triangleq \max_{i,j \in V}\{\text{dis}(i,j)\} + 1$, and $H_M(K) \triangleq D_G(M(K)+1)$. Furthermore, the state error of the system (1) satisfies following theorem.

**Theorem 1.** *Under Assumptions 1 and 2, for the constant* $\tilde{c}$, *there exist an integer* $K_1 > 0$ *such that the state error of system* (1) *satisfies*

$$|x_i(K) - x_j(K)| = O\left(\left(1 - \tilde{c}^{H_M(K)}\right)^{\frac{K}{H_M(K)}}\right) \tag{5}$$

*for all* $i, j \in V$ *and all integers* $K \geq K_1$.

The proof of Theorem 1 is given in Appendix A.

For system (1), Theorem 1 shows that the convergence rate of the state error and the growth rate of the maximum delay $M(K)$ are related. Since the relationship between the maximum delay $M(K)$ and the maximum communication distance is linear, the rate of states achieving a consensus and the growth rate of the maximum communication distances are related. It is worth noting that when the topology of system (1) satisfies Assumption 1, system (1) cannot achieve a consensus if the delay only satisfies Assumption 2. Next, in

order to make system (1) satisfy Definition 1, the growth rate of $M(k)$ needs to be reasonably restricted by the following assumption, which also is the constraint on the growth rate of the maximum communication distance.

**Assumption 3.** *For the constant $\tilde{c}$, there exists an integer $K_2 > 0$, such that*

$$\left(1 - \tilde{c}^{H_M(K)}\right)^{\frac{K}{H_M(K)}} = O(f(K)) \tag{6}$$

*for all integers $K \geq K_2$, where $f(K) = o\left(\frac{1}{K}\right)$ as $K \to \infty$.*

The following theorem shows the sufficient condition for the maximum delay $M(k)$ to satisfy Assumption 3.

**Theorem 2.** *For the constant $\tilde{c}$, any $a \in (0, \tilde{c})$, any $b \in (0, \frac{1}{2 \cdot DG})$, and any constant $M$, if $M(k) \leq -\log_a k^b + M$ for all integers $k \geq 0$, then for the constant $\tilde{c}$ and any $\epsilon \in (\hat{h}, 1)$, there exists an integer $K_2 > 0$ such that*

$$\left(1 - \tilde{c}^{H_M(K)}\right)^{\frac{K}{H_M(K)}} < \epsilon^{\sqrt{k}} \tag{7}$$

*for all $k \in Z \bigcap [K_2, \infty)$, which means that the maximum delay $M(k)$ satisfies Assumption 3.*

In Appendix B, the proof of Theorem 2 and $\hat{h}$ are given.

In the multi-agent system (1) with increasing communication distances, since the relationship between the maximum delay and the maximum communication distance is linear, if the growth rate of the maximum communication distance is not faster than the growth rate of the function $-\log_a k^b + M$, the maximum delay $M(k)$ of system (1) satisfies conditions in Theorem 2. When the maximum delay $M(k)$ satisfies Assumptions 2 and 3, the following theorem is given.

**Theorem 3.** *Under Assumptions 1–3, system (1) achieves a consensus. Furthermore, for the constant $\tilde{c}$, there exist an integer $\hat{K} > 0$, a constant $x^* \in [\min_{i \in V}\{x_i(0)\}, \max_{i \in V}\{x_i(0)\}]$, and a constant $a \in (0, 1)$, which is related to $M(\hat{K})$ and $\tilde{c}$, such that*

$$|x_i(k) - x^*| = \begin{cases} O\left(a^k\right) & k \in Z \bigcap [0, \hat{K}) \\ O(f(K)) & k \in Z \bigcap [\hat{K}, \infty) \end{cases} \tag{8}$$

*for all $i \in V$.*

For the multi-agent system (1) with increasing communication distances, Theorems 2 and 3 show that when the growth rate of the maximum communication distance is not faster than the growth rate of the function $-\log_a k^b + M$, system (1) achieves a consensus if the Assumption 1 for topology is satisfied. The proof of Theorem 3 is given in Appendix C.

Combing (5), (6), and (8), we have that under Assumptions 1–3, for all integers $k \geq \hat{K}$, the convergence rate of the consensus system (1) is equal to the rate of $\left(1 - \tilde{c}^{H_M(K)}\right)^{\frac{k}{H_M(K)}}$ tending to 0. When $k$ tends to infinity, the faster $M(k)$ tends to infinity, the slower $\left(1 - \tilde{c}^{H_M(K)}\right)^{\frac{k}{H_M(K)}}$ tends to 0. Hence, we directly obtain that the convergence rate of the consensus system (1) and the growth rate of the maximum delay $M(k)$ are negatively correlated. In other words, for the multi-agent system (1) with increasing communication distances, the convergence rate of the consensus system (1) and the growth rate of the maximum communication distance are negatively correlated.

**Theorem 4.** *Under Assumptions 1–3, for the constant $\tilde{c}$, there exist an integer $\hat{K} > 0$ such that for all integers $k \geq \hat{K}$, the rate of system (1) achieving a consensus and the rate of the maximum delay $M(k)$ tending to infinity are negatively correlated.*

Next, we use some corollaries to support Theorem 4. When the maximum time delay satisfies some special forms, we have following corollaries. The first form is the bounded maximum time delay. In other words, the maximum communication distance is bounded. Actually, Theorem 3 is suitable to consensus systems with bounded delays. For the bounded time delay $m_{ij}(K)$, its maximum time delay satisfies $M(K) \leq M$ for all integers $K \geq 0$. Then $K - m_{ij}(K) \to \infty$ as $K \to \infty$, and, for all integers $K \geq 0$, $\left(1 - \tilde{c}^{H_M(K)}\right)^{\frac{K}{H_M(K)}} = O\left(a^K\right)$, where $a = \left(1 - \tilde{c}^{DG(M+1)}\right)^{\frac{1}{DG(M+1)}} \in (0,1)$. In other words, system (1) with the bounded maximum delay $M(k)$ satisfies Assumptions 2 and 3 at the same time. As direct consequences of Theorem 3, we have the following corollary.

**Corollary 1.** *(Exponential Consensus) Suppose that $M(k) \leq M$ for all integers $k \geq 0$, where $M > 0$ is a constant integer. Under Assumption 1, system (1) achieves a consensus. Furthermore, for the constant $\tilde{c}$, there exist constants $a = \left(1 - \tilde{c}^{DG(M+1)}\right)^{\frac{1}{DG(M+1)}} \in (0,1)$ and $x^* \in [\min_{i \in V}\{x_i(0)\}, \max_{i \in V}\{x_i(0)\}]$ such that*

$$|x_i(k) - x^*| = O\left(a^k\right) \tag{9}$$

*for all $i \in V$ and all integers $k \in [0, \infty)$.*

The second form is the unbounded maximum time delay. Let $M(k) = \lfloor -\log_a k^b + M \rfloor$ for all integers $k \geq 0$, where $a \in (0, \tilde{c})$, $b \in (0, \frac{1}{2 \cdot DG})$, and $M$ is a constant. In other words, the growth rate of the maximum communication distance is not faster than the growth rate of the function $\lfloor -\log_a k^b + M \rfloor$. Obviously, the maximum time delay $M(k)$ is unbounded and the time delay $m_{ij}(k)$ corresponding to $M(k)$ satisfies Assumption 2. Furthermore, Theorem 2 shows that the unbounded maximum time delay $M(k) = \lfloor -\log_a k^b + M \rfloor$ satisfies Assumption 3. Then, combining Theorems 2 and 3, we directly have the following corollary.

**Corollary 2.** *(Asymptotical Consensus) Suppose that $M(k) = \lfloor -\log_a k^b + M \rfloor$ for all integers $k \geq 0$, where $a \in (0, \tilde{c})$, $b \in (0, \frac{1}{2 \cdot DG})$, and $M$ is a constant. Under Assumption 1, system (1) achieves a consensus. Furthermore, for the constant $\tilde{c}$ and any $\epsilon \in (\hat{h}, 1)$ shown in Lemma A2, there exist an integer $\hat{K} = \max\{K_1, K_2\}$, a constant $x^* \in [\min_{i \in V}\{x_i(0)\}, \max_{i \in V}\{x_i(0)\}]$, and a constant $a_1 \in (0,1)$ related to $M(\hat{K})$ and $\tilde{c}$ such that*

$$|x_i(k) - x^*| = \begin{cases} O\left(a_1^k\right) & k \in Z \cap [0, \hat{K}) \\ O\left(\epsilon^{\sqrt{k}}\right) & k \in Z \cap [\hat{K}, \infty) \end{cases} \tag{10}$$

*for all $i \in V$.*

When the maximum time delay is bounded, there exist an integer $\hat{K}$ such that for all integers $k \geq \hat{K}$, the growth rate of the maximum time delay is 0. Furthermore, Corollary 1 shows that the convergence rate of system (1) with bounded maximum time delays is exponential. For the unbounded maximum time delay $M(k) = \lfloor -\log_a k^b + M \rfloor$, the growth rate of the unbounded maximum delay is faster than the growth rate of the bounded maximum time delay, obviously. On the contrary, Corollary 2 shows that the convergence rate of system (1) with the unbounded maximum delay $M(k) = \lfloor -\log_a k^b + M \rfloor$ is asymptotic which is slower than the convergence rate of system (1) with bounded maximum time delays. Hence, Corollary 1 and Corollary 2 support the result of Theorem 4.

## 4. Numerical Simulation

Under the digraph $G$ with 8 agents as shown in Figure 1, numerical simulations of system (1) are executed. In Figure 1, for any $i, j \in \{1, \ldots, 8\}$, if $(i, j) \in E$, then there is a black arrow from the agent $i$ pointing to the agent $j$.

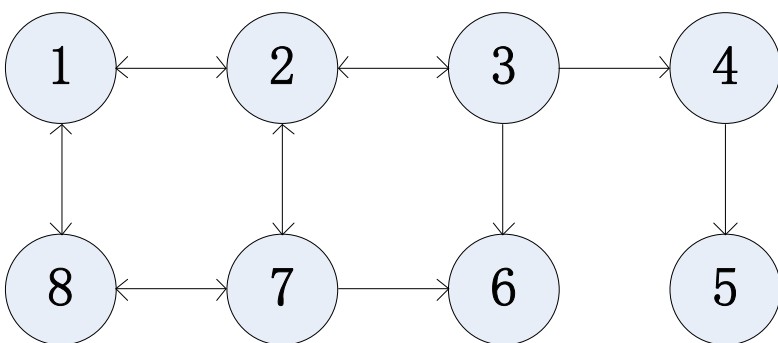

**Figure 1.** The topology graph $G$ with 8 agents.

Figure 1 shows that the digraph $G$ not only contains a spanning tree but also contains a cycle and the adjacency matrix of the digraph $G$ is partible. Hence, the digraph $G$ do not meet the topological assumptions in [25,26]. The communication cycles of agents are all $T = 1$ s. When the state of the agent $i$ is updated for the $k$th time, the distance from the center of mass of the agent $i$ to the starting point is $D_i(k)$, and $C_{ji}(k)$ represents the communication distance from the agent $i$ to the agent $j$, the maximum communication distance between agents is $C_{max}(k) = \max_{i,j \in V, s \leq k}\{C_{ji}(s)\}$. The speed of information dissemination is $v(m/s)$. In this paper, it is assumed that the relationship between the time delay and the communication distance is linear, so let $M(k) = C_{max}(k)/v$. For multi-agent systems with increasing communication distances (1), by numerically simulating the convergence of the system (1) under four different conditions where the growth rate of the communication distances is different, the conclusions of this paper are verified.

Among the four simulations, the first three simulations all use eight agents to explore a wide area from the same point, and each agent is responsible for a fan-shaped area with an angle of $\pi/4$. Record the starting point as the origin. In the simulation (1), for any integer $k \geq 0$ and any $i \in V$, $D_i^{(1)}(k) = \frac{v}{2}(-\log_a k^b)$, where $a \in (0, \tilde{c})$, $b \in \left(0, \frac{1}{2n}\right)$. Therefore, there exist $a_1 \in (0, \tilde{c})$ and $b_1 \in \left(0, \frac{1}{2n}\right)$ such that the maximum communication distance of the system satisfies $C_{max}^{(1)}(k) = v * (-\log_{a_1} k^{b_1})$. This system can be described as a system (1) with an unbounded maximum delay $M_1(k)$. In the simulation (2), for any integer $k \geq 0$ and any $i \in V$, $D_i^{(2)}(k) = \frac{vd_2 k}{2}$, where $d_2 \in (0, 1)$. Therefore, there exists $d_2' \in (0, 1)$ such that the maximum communication distance of the system satisfies $C_{max}^{(2)}(k) = vd_2'k$. The system can be described as a system (1) with an unbounded maximum delay $M_2(k)$. In the simulation (3), for any integer $k \geq 0$ and any $i \in V$, $D_i^{(3)}(k) = \frac{v}{2}(-\log_a(k^{b/2}))$. Therefore, there exist $a_3 = a_1$ and $b_3 = b_1/2$ such that the maximum communication distance of the system satisfies $C_{max}^{(3)}(k) = v * (-\log_{a_3} k^{b_3})$. The system can be described as a system (1) with an unbounded maximum delay $M_3(k)$. The simulation (4) uses 8 agents to explore in a circle with a diameter of $v(m)$. The starting points of agents are 8 octsection points on the circle, and each agent is responsible for the fan-shaped area with an angle of $\pi/4$. The agent finally converges at the center of the circle. Then, for any integer $k \geq 0$ and any $i \in V$, $D_i^{(4)}(k) = v - d_4 vk$, where $d_4 > 0$, and satisfies $D_i^{(4)}(k) \geq 0$. Therefore, the maximum communication distance of the system satisfies $C_{max}^{(4)}(k) \leq 2v$. This system can be described as a system (1) with a bounded maximum delay $M_4(k)$. It can be noticed that in the first three simulations, the maximum communication distance $C_{max}^{(1)}(k)$, $C_{max}^{(2)}(k)$ and $C_{max}^{(3)}(k)$

tends to infinity as the number of iterations $k$ growths, and the maximum communication distance $C_{max}^{(4)}(k)$ in the simulation (4) is bounded.

Figure 2 describes the motion trajectories of eight agents in four simulations, in which the abscissa and ordinate describe the position of any point in the two-dimensional plane, and the unit is $m$. The colored curves are the motion trajectories of all agents. Starting from the purple trajectory and rotating clockwise to the green trajectory, corresponding agent numbers are $1, 2, 3, 4, 5, 6, 7, 8$, respectively; and the black dashed line is used to separate the detection areas of the eight agents. In Figure 2, the circle enclosed by the black dashed line in the sub-figure simulation (4) is the boundary of the detection area in the simulation (4).

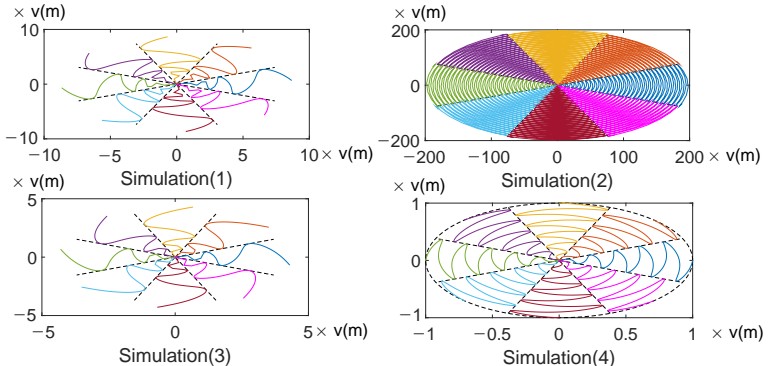

**Figure 2.** The motion trajectories of 8 agents.

Figures 3, 4, 6 and 7 show the variation of the maximum communication distances between agents $C_{max}^{(s)}(k), s = 1, 2, 3, 4$ and the maximum delays $M_s(k), s = 1, 2, 3, 4$ with the number of iterations $k$ of four simulations, respectively. It can be seen that the maximum communication distance $C_{max}(k)$ and the maximum communication delay $M(k)$ meet the same changing trend. Figure 5 is a comparison figure of change curves of the agent state value in the simulation (1) and the simulation (2). Figure 8 describes the logarithmic function curve of the state error between the agents in the simulation (1), the simulation (3) and the simulation (4) in logarithmic coordinates.

First, compare the simulation (1) and the simulation (2). Since the growth rate of $D_i^{(2)}(k)$ in the simulation (2) is faster than the growth rate of $D_i(k)^{(1)}$ in the simulation (1), Figure 2 shows that the range of agent detection in the simulation (2) obviously exceeds the range of agent detection in the simulation (1).

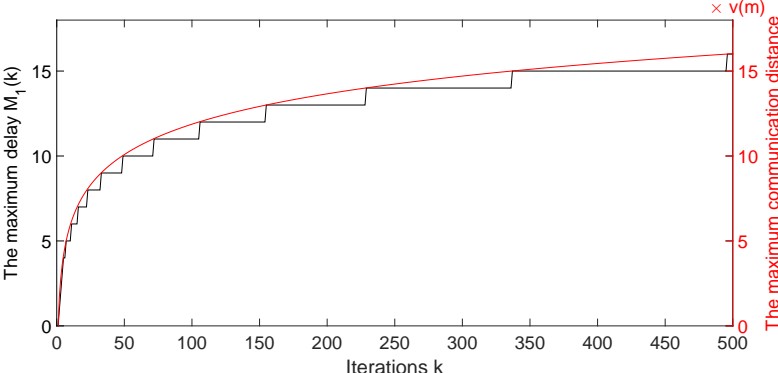

**Figure 3.** The maximum communication distances $C_{max}^{(1)}(k)$ between agents and the maximum delays $M_1(k)$ in the simulation (1).

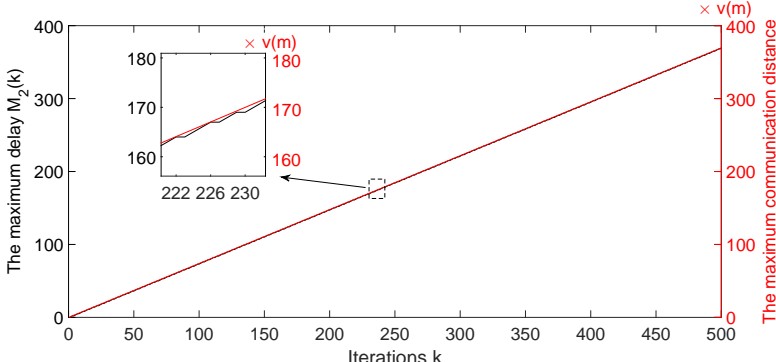

**Figure 4.** The maximum communication distances $C_{max}^{(2)}(k)$ between agents and the maximum delays $M_2(k)$ in the simulation (2).

Figure 3 shows that since the maximum communication distance satisfies $C_{max}^{(1)}(k) = v * (-\log_{a_1} k^{b_1})$, the curve of the maximum communication delay $M_1(k)$ closely follows the logarithmic function curve $g_1(k) = -\log_{a_1} k^{b_1}$. Figure 4 shows that since the maximum communication distance satisfies $C_{max}^{(2)}(k) = vd_2'k$, the curve of the maximum communication delay $M_2(k)$ closely follows the linear function curve $g_2(k) = d_2'k$. Therefore, the unbounded maximum delay $M_1(k)$ satisfies the condition in Corollary 2, and the unbounded maximum delay $M_2(k)$ only satisfies Assumption 2.

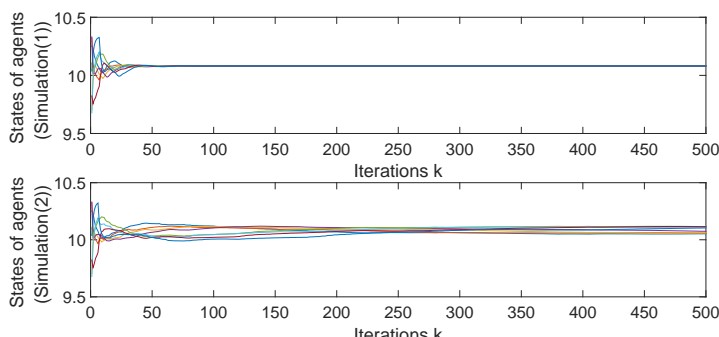

**Figure 5.** The state change curve of agents in the simulation (1) and the simulation (2).

Figure 5 shows that the system with the maximum delay $M_1(k)$ is convergent, but the system with the maximum delay $M_2(k)$ is divergent. In other words, when the growth rate of the maximum communication distance can ensure that the maximum delay satisfies both Assumption 2 and Assumption 3, the system (1) can reach a consensus under Assumption 1 for topology, which is more relaxed than the topological assumptions in [25,26].

Next, compare the simulation (1), the simulation (3) and the simulation (4). Since $D_i^{(1)}(k)$ growths faster than $D_i^{(3)}(k)$, Figure 2 shows that the detection range of agents in the simulation (1) is larger than the detection range of agents in the simulation (3). Furthermore, since $D_i^{(3)}(k)$ increases with the growth of $k$, $D_i^{(4)}(k)$ decreases with the growth of $k$, there exists $K > 0$ such that $D_i^{(3)}(k) > D_i^{(4)}(k)$ for all integers $k > K$. In other words, due to the limited detection range of the simulation (4), as long as the detection time is long enough, the detection range of agents in the simulation (1) and the simulation (3) will always be larger than the detection range of agents in the simulation (4).

Figure 6 shows that due to the maximum communication distance satisfies $C_{max}^{(3)}(k) = v * (-\log_{a_3} k^{b_3})$, the curve of the maximum communication delay $M_3(k)$ closely follows the logarithmic function curve $g_3(k) = -\log_{a_3} k^{b_3}$. Therefore, the unbounded maximum delay $M_3(k)$ satisfies the condition in Corollary 2. Figure 7 shows that due to the maximum communication distance satisfies $C_{max}^{(4)}(k) \equiv v$, the curve of the maximum communication

delay $M_3(k)$ satisfies $M_4(k) \equiv 1$, which is bounded. In other words, the unbounded maximum delay $M_4(k)$ satisfies the condition in Corollary 1. Obviously, the maximum delay in the simulation (1), the simulation (3) and the simulation (4) satisfies that the growth rate of $M_1(k)$ is greater than the growth rate of $M_3(k)$, and the growth rate of $M_3(k)$ is greater than the growth rate of $M_4(k)$.

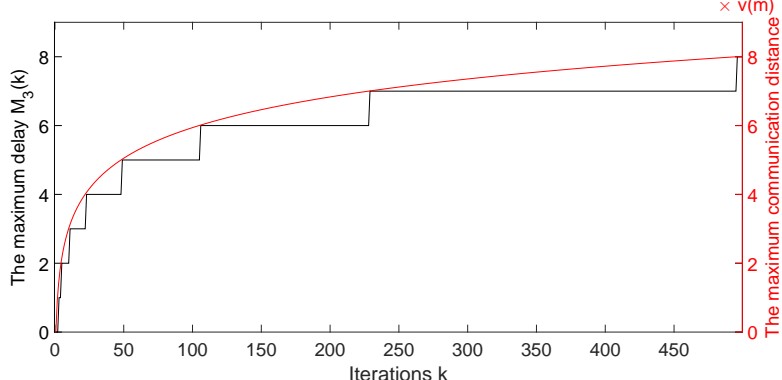

**Figure 6.** The maximum communication distances $C_{max}^{(3)}(k)$ between agents and the maximum delays $M_3(k)$ in the simulation (3).

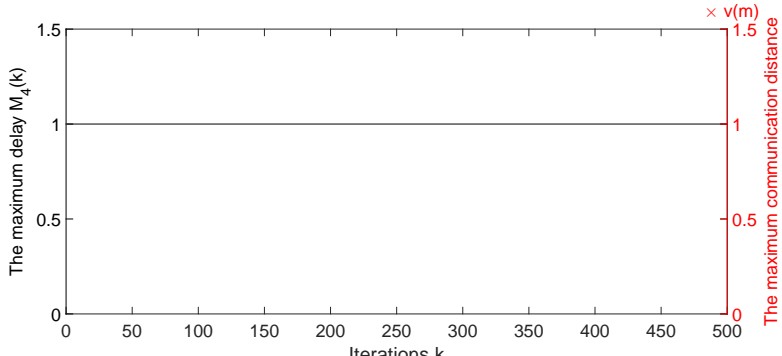

**Figure 7.** The maximum communication distances $C_{max}^{(4)}(k)$ between agents and the maximum delays $M_4(k)$ in the simulation (4).

Figure 8 describes the system error curve $F(k) = \| \max_{i,j}\{x_i(k) - x_j(k)\} \|$ in the simulation (1), the simulation (3) and the simulation (4) under logarithmic coordinates $\ln(k) - \ln(F(k))$. In Figure 8, the curve of the function $f_1(k) = 0.4^{\sqrt{k}}$ in logarithmic coordinates is a red dashed line, the curve of the function $f_3(k) = 0.1^{\sqrt{k}}$ in logarithmic coordinates is a blue dashed line, and the curve of the function $f_4(k) = 0.7^k$ in logarithmic coordinates is a black dashed line. The red solid line is the $F(k)$ of the system (1) with the unbounded maximum delay $M_1(k)$. It is below the red dashed line $f_1(k)$, indicating that the convergence rate of the system (1) with the unbounded maximum delay $M_1(k)$ is $O\left(0.4^{\sqrt{k}}\right)$, and the system is asymptotically convergent. The blue solid line is the $F(k)$ of the system (1) with the unbounded maximum delay $M_3(k)$. It is below the blue dashed line $f_3(k)$, indicating that the convergence rate of the system (1) with the unbounded maximum delay $M_3(k)$ is $O\left(0.1^{\sqrt{k}}\right)$. In other words, the system (1) with unbounded maximum delay $M_3(k)$ is asymptotically convergent and has a faster convergence rate than the convergence rate of the system (1) with unbounded maximum delay $M_1(k)$. The black solid line is the $F(k)$ of the system (1) with the unbounded maximum delay $M_4(k)$. It is below the black dashed line $f_4(k)$, indicating that the convergence rate of the system (1) with the unbounded maximum delay $M_4(k)$ is $O\left(0.7^k\right)$, which is exponential. It is faster than the convergence rate of the system (1) in simulation (1) and simulation (3). Therefore, the

comparison simulation supports the conclusion of Theorem 4 the the convergence rate of the system (1) is negatively correlated with the growth rate of the maximum delay.

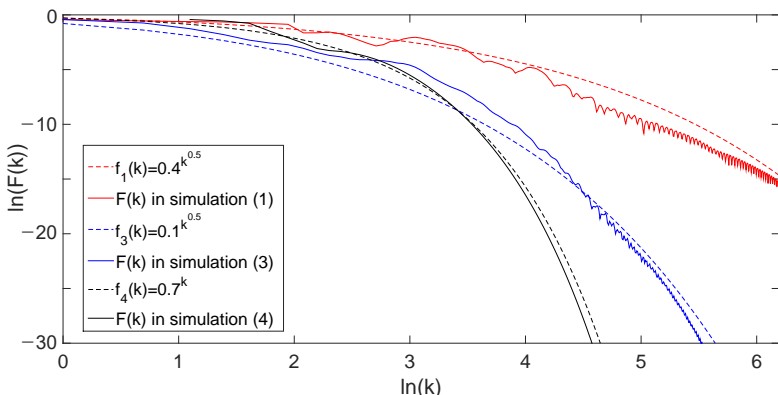

**Figure 8.** The system error curve in the simulation (1), the simulation (3) and the simulation (4) under logarithmic coordinates.

## 5. Conclusions

In this paper, the consensus problem of multi-agent systems is studied with non-distributed unbounded time-delays caused by the growth of communication distances. The multi-agent system is modeled as a discrete time-delay system. Then, for any finite integer $k > 0$, the finite-dimensional augmented model of the time-delay system is built in the interval $[0, k]$ to study the system state. Firstly, under the topology containing a spanning tree and a mild condition about the growth rate of the maximum delay, which also is a constraint on the growth rate of the maximum communication distance, the system is proved to achieve a consensus. Then, this paper shows that the rate of the system achieving a consensus and the growth rate of the maximum delay are negatively correlated. Furthermore, the rate of the system achieving a consensus and the growth rate of the maximum communication distance also satisfy the same relationship. It is worth noting that these results are applicable to any multi-agent system that can be described as the discrete system with non-distributed unbounded time-varying delays. Furthermore, in the future research, we will study the influence of the unbounded delay containing the transmutation delay and the a process scheduling delay on the modeling and consensus conditions of the system. It is foreseeable that the research results will be applicable to a wider range of practical systems.

**Author Contributions:** Conceptualization, S.Z. and Y.-P.T.; methodology, S.Z.; software, S.Z.; validation, S.Z. and Y.-P.T.; formal analysis, S.Z.; investigation, S.Z.; resources, S.Z.; data curation, S.Z.; writing—original draft preparation, S.Z.; writing—review and editing, S.Z. and Y.-P.T.; visualization, S.Z.; supervision, Y.-P.T.; project administration, Y.-P.T.; funding acquisition, Y.-P.T. All authors have read and agreed to the published version of the manuscript.

**Funding:** This research was funded by the National Natural Science Foundation of China under grant 61573105.

**Institutional Review Board Statement:** Not applicable.

**Informed Consent Statement:** Not applicable.

**Data Availability Statement:** Not applicable.

**Conflicts of Interest:** The authors declare no conflict of interest.

## Appendix A. Proof of Theorem 1

In order to prove Theorem 1, the following lemma is given.

**Lemma A1.** *Under Assumption 1, for any $P(l) \in S(K)$, $l \in \{1, \cdots, H_M(K)\}$, $\prod_{l=1}^{H_M(K)} P(l)$, the product of these $H_M(K)$ matrices, contains a column in which all entries are not zero.*

**Proof of Lemma A1.** For any $n(M(K) + 1)$-dimensional square matrix $P \in S(K)$, We divide the matrix into blocks as follows

$$
P = \begin{bmatrix} P_{[11]} & \cdots & P_{[1(M(K)+1)]} \\ \vdots & \ddots & \vdots \\ P_{[(M(K)+1)1]} & \cdots & P_{[(M(K)+1)(M(K)+1)]} \end{bmatrix},
$$

where for $i, j \in \{1, \ldots, M(K) + 1\}$, $P_{[ij]} \in R^{n \times n}$. For any $s \geq 1$ and any $P(l) \in S(K)$, $l \in \{1, \cdots, s\}$, the product of any $s$ matrices belonging to the set $S(K)$ is represented by $\prod_{l=1}^{s} P(l)$. Firstly, for $\prod_{l=1}^{M(K)+1} P(l)$ and any $i \in \{1, \ldots, M(K) + 1\}$, diagonal entries of $(\prod_{l=1}^{M(K)+1} P(l))_{[i1]}$ are all non-zero. Under Assumption 1, without loss of generality, let the agent 1 be the root agent. There must exists $V_1 \subset V$ such that $(1, s_1) \in E$ for all $s_1 \in V_1$. Hence, for any $P(M(K) + 2) \in S(K)$ and any $s_1 \in V_1$, $(\prod_{l=1}^{M(K)+2} P(l))_{[11]})_{s_1 1}$ is non-zero. Then, for $\prod_{l=1}^{2(M(K)+1)} P(l)$, any $i \in \{1, \ldots, M(K) + 1\}$, and any $s_1 \in V_1$, diagonal entries and $s_1 - 1$ entry of $(\prod_{l=1}^{2(M(K)+1)} P(l))_{[i1]}$ are all non-zero. There also exists $V_2 \subset V$, which satisfies that for each $s_2 \in V_2$, and there is at least one $s_1 \in V_1$ such that $(s_1, s_2) \in E$. Hence, for any $P(2M(K) + 3) \in S(K)$ and any $s \in V_1 \cup V_2$, $((\prod_{l=1}^{2M(K)+3} P(l))_{[11]})_{s1}$ is non-zero. Then, for $\prod_{l=1}^{3(M(K)+1)} P(l)$, any $i \in \{1, \ldots, M(K) + 1\}$, and any $s \in V_1 \cup V_2$, diagonal entries and $s - 1$ entry of $(\prod_{l=1}^{3(M(K)+1)} P(l))_{[i1]}$ are all non-zero. Since subscripts of sets $V_r$ all satisfy the inequality that $r \leq D_G - 1$, and Assumption 1 is satisfied, we have that $\{1\} \cup V_1 \cup \cdots \cup V_{D_G-1} = V$. Therefore, for any $P(l) \in S(K)$, $l \in \{1, \cdots, H_M(K)\}$, $\prod_{l=1}^{H_M(K)} P(l)$ contains a column in which all entries are not zero. $\square$

Based on Lemma A1, the proof of Theorem 1 is given.

**Proof of Theorem 1.** For any integer $K \geq 1$, we have (3) for all $k \in Z \bigcap [0, K - 1]$. For any integers $k_1, k_2$ satisfying $0 \leq k_1 \leq k_2 < K$, let $R_K(k_2, k_1) = \prod_{s=k_1}^{k_2} A_K(s)$. It can be noticed that $R_K(k_1, k_1) = I$. According to Assumption 2, there exists an integer $K_1 > 0$ such that $K > H_M(K)$ for all $K \in Z \bigcap [K_1, \infty)$. Under Assumption 1, Lemma A1 shows that for any integer $K \in [K_1, \infty)$ and all integers $k \in [H_M(K), K - 1]$, $R_K(k, k - H_M(K))$ contains a column in which all entries are not zero. Hence, for any integer $K \geq K_1$, we have that $\max_j \min_i \{R_K(k, k - H_M(K))_{ij}\} \geq \tilde{c}^{H_M(K)} > 0$ for all integers $k \in [H_M(K), K - 1]$. We consider the Lyapunov function $V(y_{x,K}(k)) = \max_i \{(y_{x,K})_i(k)\} - \min_i \{(y_{x,K})_i(k)\}$. Applying Lemma 2 in [3], we get that for any integer $K \geq K_1$ and all integers $k \in [0, K]$, the function $V(y_{x,K}(k))$ satisfies the following inequality that

$$
\begin{aligned}
V(y_{x,K}(k)) &\leq \left(1 - \max_j \min_i \{R_K(k-1, k - H_M(K) - 1)_{ij}\}\right) \\
&\quad \cdot V(y_{x,K}(k - H_M(K) - 1)) \\
&\leq \left(1 - \tilde{c}^{H_M(K)}\right) V(y_{x,K}(k - H_M(K) - 1)) \\
&\leq \left(1 - \tilde{c}^{H_M(K)}\right)^{\left\lfloor \frac{k}{H_M(K)} \right\rfloor} V(y_{x,K}(0)) \\
&= \left(1 - \tilde{c}^{H_M(K)}\right)^{\left\lfloor \frac{k}{H_M(K)} \right\rfloor} \max_{i,j \in V} \{|x_i(0) - x_j(0)|\}.
\end{aligned}
\tag{A1}
$$

When $k = K$, Equation (A1) implies that for any integer $K \geq K_1$,

$$|x_i(K) - x_j(K)| \leq V(y_{x,K}(K)) \leq O\left( \left( 1 - \tilde{c}^{H_M(K)} \right)^{\frac{K}{H_M(K)}} \right). \tag{A2}$$

$\square$

### Appendix B. Proof of Theorem 2

For proving Theorem 2, the following lemma is given.

**Lemma A2.** *If $M(k)$ satisfies that for a $\mathcal{K}_\infty$ class function $f(x) = \left( (D_G x) \tilde{c}^{-D_G x} \right)^2$ and the constant $\tilde{c}$, there exist $K_3 \in Z \cap [0, \infty)$ such that $M(k) + 1 < f^{-1}(k)$ for all $k \in Z \cap [K_3, \infty)$, then for the constant $\tilde{c}$ and any $\epsilon \in (\hat{h}, 1)$, there exists an integer $K_2 \geq K_3$ such that*

$$\left( 1 - \tilde{c}^{D_G(M(k)+1)} \right)^{\frac{k}{D_G(M(k)+1)}} < \epsilon^{\sqrt{k}} \tag{A3}$$

*for all $k \in Z \cap [K_2, \infty)$, where $\hat{h}' = \hat{h}^{\sqrt{\delta}}$, and $\hat{h} \in [0, \frac{1}{e}]$ is shown by Lemma 2 in [27].*

**Proof of Lemma A2.** Firstly, according to $M(k) + 1 < f^{-1}(k)$ for all $k \in Z \cap [K_3, \infty)$, we get that $\frac{k}{f(M(k)+1)} > 1$ for all $k \in Z \cap [K_3, \infty)$. Then, we have

$$\frac{k}{D_G(M(k) + 1)} > \frac{\sqrt{k f(M(k))}}{D_G(M(k) + 1)} \geq \sqrt{k} \tilde{c}^{-D_G(M(k)+1)}$$

for all $k \in Z \cap [K_3, \infty)$.

Secondly, since $M(k) \to \infty$ as $k \to \infty$, we can get from Lemma 2 in [27] that for any $\epsilon \in (\hat{h}, 1)$, there exists an integer $K_3' > 0$ such that $(1 - \tilde{c}^{D_G(M(k)+1)})^{\tilde{c}^{-D_G(M(k)+1)}} < \epsilon$ for all $k \in Z \cap [K_3', \infty)$, where $K_3'$ is related to $\epsilon$ and $\tilde{c}$.

Hence, for any $\epsilon \in (\hat{h}, 1)$, there exists $K_2 = \max\{K_3, K_3'\} > 0$ such that

$$\left( 1 - \tilde{c}^{D_G(M(k)+1)} \right)^{\frac{k}{D_G(M(k)+1)}} < \epsilon^{\sqrt{k}}$$

for all $k \in Z \cap [K_2, \infty)$. $\square$

Now, the proof of Theorem 2 is given.

**Proof of Theorem 2.** For the $\mathcal{K}_\infty$ class function $f(x) = \left( (D_G x) \tilde{c}^{-D_G x} \right)^2$, since $a \in (0, \tilde{c})$, we have

$$f(M(k) + 1) \leq \left( (D_G(-\log_a k^b + M + 1)) \tilde{c}^{-D_G(-\log_a k^b + M + 1)} \right)^2$$

$$< \left( (D_G(-\log_a k^b + M + 1)) \tilde{c}^{-D_G(M+1)} \right)^2 k^{2bD_G}.$$

Furthermore, since $b \in (0, \frac{1}{2 \cdot D_G})$, we have that $2bD_G < 1$. Hence, we have that there exists an integer $K_3 > 0$ such that $f(M(k) + 1) < k$ for all integers $k \geq K_3$. Since $f(x)$ is the $\mathcal{K}_\infty$ class function, for all integers $k \geq K_3$, we have that $M(k) + 1 < f^{-1}(k)$. Then, using Lemma A2, we have that there exists a $K_2 \geq K_3$ such that the unbounded maximum time delay $M(k) \leq -\log_a k^b + M$ satisfies (A3) for all $k \in Z \cap [K_2, \infty)$. $\square$

### Appendix C. Proof of Theorem 3

In this section, the proof of Theorem 3 is given.

**Proof of Theorem 3.** Lyapunov function $V(y_{x,K}(k)) = \max_i\{(y_{x,K})_i(k)\} - \min_i\{(y_{x,K})_i(k)\}$ is considered. According to (A2) and Assumption 3, there exist an integer $\hat{K} = \max\{K_1, K_2\}$ such that

$$|x_i(K) - x_j(K)| = O(f(K)) \tag{A4}$$

for any integer $K \geq \hat{K}$. Furthermore, according to (A1), there exists a constant $a \in (0,1)$ being related to $M(\hat{K})$ and $\tilde{c}$ such that

$$|x_i(k) - x_j(k)| \leq V(y_{x,\hat{K}}(k)) \leq \mathrm{O}\left(a^k\right) \tag{A5}$$

for all integers $k \in [0, \hat{K}]$.

From (A4) and (A5), it follows that

$$|x_i(K) - x_j(K)| \to 0, K \to \infty. \tag{A6}$$

From (3), we have

$$x_1(K) = x_1(K-1) + \left( \sum_{s=0}^{M(K)} \sum_{i \in V} \left( [\widehat{A_{Ks}}(K-1)]_{1i} x_i(K-1-s) \right) - x_1(K-1) \right) \tag{A7}$$

for all integers $K \geq 1$. Because of

$$\frac{\left(1 - \tilde{c}^{H_M(K)}\right)^{\frac{K-1}{H_M(K)}}}{\left(1 - \tilde{c}^{H_M(K)}\right)^{\frac{K}{H_M(K)}}} \to 1 \ as \ K \to \infty,$$

then

$$\left(1 - \tilde{c}^{H_M(K)}\right)^{\frac{K-1}{H_M(K)}} = \mathrm{O}\left( \left(1 - \tilde{c}^{H_M(K)}\right)^{\frac{K}{H_M(K)}} \right) \tag{A8}$$

for all integers $K \in [\hat{K}, \infty)$. From (A1), (6) shown in Assumption 3, and (A8), the function $V(y_{x,K}(K-1))$ satisfies the following inequality that

$$V(y_{x,K}(K-1)) = \mathrm{O}(f(K)) \tag{A9}$$

for all integer $K \in [\hat{K}, \infty)$.

According to (A9), we have

$$\max_{i \in V, K-M(K)-1 \leq k \leq K-1} |x_1(K-1) - x_i(k)| \leq V(y_{x,K}(K-1))$$
$$= \mathrm{O}(f(K)) \tag{A10}$$

for all integer $K \geq \hat{K}$.

Then, combining (A7) and (A10), we have

$$|x_1(K) - x_1(K-1)| \leq \mathrm{O}(f(K)) \tag{A11}$$

for all integer $K \geq \hat{K}$.

According to (A11) and $f(K) = \mathrm{o}\left(\frac{1}{K}\right)$ as $K \to \infty$ shown in Assumption 3, by using the Cauchy's convergence test, we can get that there exists a constant $x^*$ such that $|x_i(k) - x^*| \to 0, k \to \infty$ for all $i \in V$. Since there are no external inputs in system (1), we have $x^* \in [\min_{i \in V}\{x_i(0)\}, \max_{i \in V}\{x_i(0)\}]$. Therefore, system (1) satisfies Definition 1. $\square$

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
