# Peer review of "Consensus of Multi-Agent Systems with Unbounded Time-Varying Delays"

_applsci, doi:10.3390/app11114944_

Round 1

Reviewer 1 Report

Comments to the authors :

The paper analyzes the convergence of a multi-agent distributed consensus problem under unbounded time delays. The analysis is based on three key assumptions on the communication delay and the growth rate of the delay. The paper proves, if the multi-agent system contains a spanning tree and the growth rate of the delay is bounded by the growth rate of the maximum communication distance, the system achieves consensus. 

The paper has the following key strengths. It looks into finding a weak set of requirements of the interconnected network topology under unbounded time delay. The theorems and the proves in the paper are mathematically sound. The related work section includes necessary references to the prior work on which the authors' contribution is based on.

However, the paper has the following key questions which the authors have not adequately addressed. One of the main assumptions of the paper is that the communication delay in a system grows linearly with the inter-agent "communication distance". However, many of the real-world robots ( which are either operated on air or on land ) use radio signals to communicate. Marine robots use sonar to communicate instead of radio signals. Hence, the delay in communication and the distance between the robots are hardly co-related in practice. Therefore, the authors have failed to provide proper motivation and relevance to a real-world problem to justify the theoretical study on delay. ( Communication delay can either be a transmutation delay where an agent has to transmit multiple messages due to packet loss/failures of the communication devices, or it can be a process scheduling delay as the received messages are not processed in time because the controller process is preempted by processes with a higher priority ). The reviewer suggests the authors re-model the research problem to be close to a real-world multi-agent system. 

The minor weaknesses are that the paper includes a broken pointer to reference ( line 411 ). The authors do not adequately explain why the agent's own state is discounted by a factor of 1-c in Assumption 1.  

Author Response

Response to Reviewer 1 Comments

Point 1: (1) The paper has the following key questions which the authors have not adequately addressed. One of the main assumptions of the paper is that the communication delay in a system grows linearly with the inter-agent "communication distance". However, many of the real-world robots (which are either operated on air or on land) use radio signals to communicate. Marine robots use sonar to communicate instead of radio signals. Hence, the delay in communication and the distance between the robots are hardly co-related in practice. Therefore, the authors have failed to provide proper motivation and relevance to a real-world problem to justify the theoretical study on delay. (2) ( Communication delay can either be a transmutation delay where an agent has to transmit multiple messages due to packet loss/failures of the communication devices, or it can be a process scheduling delay as the received messages are not processed in time because the controller process is preempted by processes with a higher priority ). The reviewer suggests the authors re-model the research problem to be close to a real-world multi-agent system.

Response 1: (1) Thanks for your suggestion. As you said, in the previous version, we did not provide a more in-depth explanation of the significance of this research and the reason why we only consider the delay caused by the communication distance. The reason why we studied the influence of the delay caused by the communication distance on the consensus of the multi-agent system was mainly based on the following considerations. In order to perform precise tasks, the state between agents needs to be exactly synchronized. For example, in the field of the military reconnaissance, if a distributed multi-agent system needs to cooperatively observe targets, the agents need to achieve exact time synchronization. In this type of systems, the time synchronization error between agents needs to reach the microsecond or nanosecond level, and the smaller the time synchronization error is, the better the cooperative task of agents is completed. In this case, although the delay caused by the communication distance between agents is very small, which may only reach the millisecond or microsecond level, it still cannot be ignored. And when the number of agents in the system is small and the distance between agents is far, the density of the network is small. In this kind of network, the delay caused by the packet loss and the channel congestion decreases. On the contrary, the delay caused by the communication distance increases and has a greater impact on the communication time-delay.

(2) Thanks for your suggestion. As you said, although the delay containing the transmutation delay and the process scheduling delay is indeed widespread in practical multi-agent systems, this article does not consider such delay. Your suggestion made us notice that by studying the modeling and consensus problems of unbounded delay systems with such time delay, the conclusions can be applied to more practical systems. This is undoubtedly interesting and challenging. Therefore, we will take this issue as the direction of our further research.

For (1): In the revised version, in the second paragraph of Introduction, we introduced the reasons of studying the influence of the time delay caused by the communication distance on the system and added relevant references, so that the research of this article has a clearer practical significance.

For (2): In the revised version, we added instructions for further research directions at the end of the conclusion.

Point 2: (1)The paper includes a broken pointer to reference ( line 411 ). (2)The authors do not adequately explain why the agent's own state is discounted by a factor of 1-c in Assumption 1.

Response 2: Thanks for your suggestion. (1) As you said, in the previous version, the pointer to reference in line 411 is broken. (2)  As you said, in the previous version, we do not adequately explain why the agent's own state is discounted by a factor of 1-c.

For (1): In the revised version, the pointer to reference in line 413 has been repaired.

For (2): In the revised version, we show that the parameter c is a designable constant gain of the system (1) (line 184). At the same time, the form of system (1) is rewritten to make the meaning of parameter c clearer (line 183).  

The corresponding modification has been marked in red in the attachment.

Reviewer 2 Report

The paper has merit. It brought out some interesting results related to characterizing the effect of unbounded delays. However, it needs extensive editing.  Please see attached document that highlights some of the places where editing is required.

Author Response

Response to Reviewer 2 Comments

Point 1: In the previous version, there were a highlighted syntax error (As is well known) and a highlighted semantic repetition (is often encountered) in line 28.

Response 1: Thanks for your suggestion. In the revised version, “As is well known, delay is unavoidable in many practical systems and is often encountered, which……” is rewritten as “It is a well-known fact that delay is unavoidable in many practical systems, which……” in line 26.

Point 2:  In the previous version, there was an unclear expression (Since the difference of the discrete Lyapunov-Krasovskii functional contains the positive quadratic term whose coefficient is the time-delay, the boundedness of the time-delay is the necessary condition for the existence of the condition that the difference is negative definite.) in line 50.

Response 2: Thanks for your suggestion. In the revised version, “Since the difference of the discrete Lyapunov-Krasovskii……the difference is negative definite.” is rewritten as “In the difference of the discrete Lyapunov-Krasovskii functional, there is a positive quadratic term and its coefficient is the time-delay. Therefore, only if the time-delay is bounded, conditions that makes the difference negative definite exist.” in line 65.

Point 3: In the previous version, there were highlighted syntax errors (facing) in line 68 and (strong) in line 80.

Response 3: Thanks for your suggestion. In the revised version, “facing” is rewritten as “faces” in line 83. And “strong” is rewritten as “strongly” in line 93.

Point 4: In the previous version, there was an unclear expression (Since the maximum delay and the maximum communication distance are positively correlated, the mild assumption is the constraint on the rate of the maximum communication distance between agents, obviously.) in line 123.

Response 4: Thanks for your suggestion. In the revised version, “Since the maximum delay ……distance between agents, obviously.” is rewritten as “Since the maximum delay and the maximum communication distance are positively correlated, the assumption of the growth rate of the maximum delay also is the constraint on the growth rate of the maximum communication distance.” in line 128.

Point 5: In the previous version, there was an unclear expression (And the agent updates its own state with the state of a neighbor received last time, if the agent does not receive the state sent by the neighbor in the period T.) in line 166.

Response 5: Thanks for your suggestion. In the revised version, “And the agent updates its own state with the state of a neighbor …… state sent by the neighbor in the period T.” is rewritten as “If an agent does not receive the state information sent by one of its neighbors in a period T, then the agent updates its own state with the last received state information of the neighbor.” in line 169.

Point 6:  In the previous version, the definition of c was not given in line 176.

Response 6: Thanks for your suggestion. In the revised version, the definition of c is given as “c∈(0, 1) is a designable constant gain of system (1)” in line 184.

Point 7:  In the previous version, “In (1), if (i,j)∈E , then aji >0 . Otherwise, aji =0. And we have \sum_{i=1}^n aji =1” in line 177 needed be rewritten concisely.

Response 7: Thanks for your suggestion. In the revised version, “In (1), if (i,j)∈E , then aji >0 . Otherwise, aji =0. And we have \sum_{i=1}^n aji =1.” is rewritten as “In (1), \sum_{i=1}^n aji =1, where  aji >0  when (i,j)∈E, and  aji =0 otherwise.” in line 185.

Point 8:  In the previous version, there was an unclear expression (And under Assumption 1, Assumption 2 does not guarantee that system (1) achieves a consensus.) in line 226.

Response 8: Thanks for your suggestion. In the revised version, “And under Assumption 1, Assumption 2 does not guarantee that system (1) achieves a consensus.” is rewritten as “It is worth noting that when the topology of system (1) satisfies Assumption 1, system (1) cannot achieve a consensus if the delay only satisfies Assumption 2.” in line 228.

The corresponding modification has been marked in red in the attachment.
